# Fluid Language Model Benchmarking

**Valentin Hofmann**◇♡   **David Heineman**◇   **Ian Magnusson**◇♡   **Kyle Lo**◇
**Jesse Dodge**◇   **Maarten Sap**◇♠   **Pang Wei Koh**◇♡   **Chun Wang**♡
**Hannaneh Hajishirzi**◇♡   **Noah A. Smith**◇♡
◇Allen Institute for AI   ♡University of Washington   ♠Carnegie Mellon University

## Abstract

Language model (LM) benchmarking faces several challenges: comprehensive evaluations are costly, benchmarks often fail to measure the intended capabilities, and evaluation quality can degrade due to labeling errors and benchmark saturation. Although various strategies have been proposed to mitigate these issues, they tend to address individual aspects in isolation, neglecting broader questions about overall evaluation quality. Here, we introduce FLUID BENCHMARKING, a new evaluation approach that advances LM benchmarking across multiple dimensions. Inspired by psychometrics, FLUID BENCHMARKING is based on the insight that the relative value of benchmark items depends on an LM's capability level, suggesting that evaluation should adapt to each LM. Methodologically, FLUID BENCHMARKING estimates an *item response model* based on existing LM evaluation results and uses the inferred quantities to *select evaluation items dynamically*, similar to computerized adaptive testing in education. In our experiments, we compare FLUID BENCHMARKING against the common practice of random item sampling as well as more sophisticated baselines, including alternative methods grounded in item response theory. We examine four dimensions—efficiency, validity, variance, and saturation—and find that FLUID BENCHMARKING achieves superior performance in all of them (e.g., higher validity *and* less variance on MMLU with fifty times fewer items). Our analysis shows that the two components of FLUID BENCHMARKING have distinct effects: item response theory, used to map performance into a latent ability space, increases validity, while dynamic item selection reduces variance. Overall, our results suggest that LM benchmarking can be substantially improved by moving beyond static evaluation.

 **Code and Data**   github.com/allenai/fluid-benchmarking

## 1 Introduction

The field of language model (LM) evaluation is experiencing a moment of crisis. With new benchmarks being released by the day, it becomes increasingly difficult to decide which benchmark(s) to pick for a certain evaluation goal (Ni et al., 2024; Perlitz et al., 2024b). At the same time, evaluating LMs on ever-growing sets of benchmarks leads to substantial computational—and, consequently, financial and environmental—costs (Liang et al., 2023), all while producing brittle results that fluctuate due to evaluation noise (Madaan et al., 2024; Mizrahi et al., 2024). More alarmingly, it is often unclear whether a specific benchmark in fact measures the capability that it purports to evaluate (Liao et al., 2021; Saxon et al., 2024), a problem exacerbated by labeling errors (Northcutt et al., 2021; Gema et al., 2024; Vendrow et al., 2025) and benchmark saturation, when many LMs are scoring near the maximum on a benchmark (Vania et al., 2021; Xia et al., 2024).

These challenges have spurred various efforts to improve benchmarking, by increasing efficiency (Perlitz et al., 2024a; Polo et al., 2024; Vivek et al., 2024; Kipnis et al., 2025), detecting and correcting mislabeled items (Gema et al., 2024; Vendrow et al., 2025), reducing evaluation variance (Madaan et al., 2024), and enhancing benchmark difficulty (Suzgun

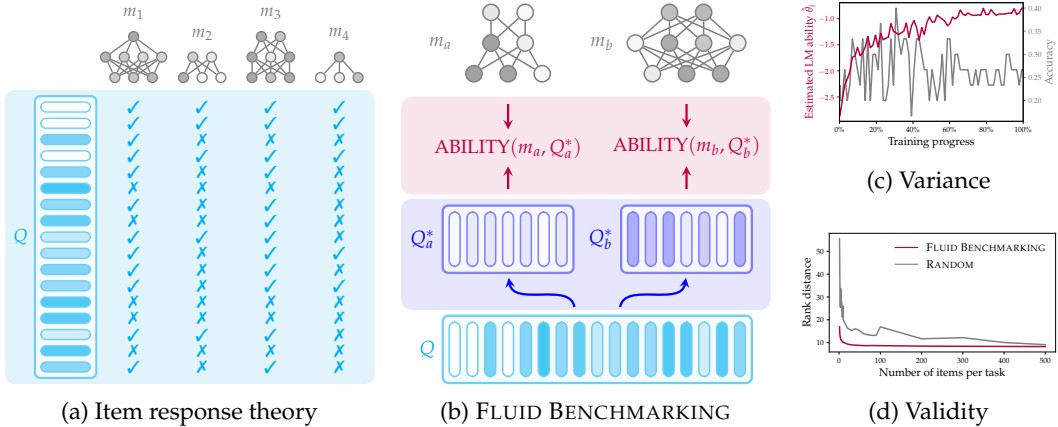

(a) Item response theory    (b) Fluid Benchmarking    (c) Variance / (d) Validity

Figure 1: (a) Given a benchmark $Q$, we train an IRT model on publicly available LM evaluation results, providing useful information about individual items (specifically, about difficulty and discrimination). The figure illustrates this with results for four LMs and difficulty, symbolized by item darkness. In practice, we use more than a hundred LMs. (b) Fluid Benchmarking leverages the IRT-enriched benchmark in two ways: it uses item difficulty and discrimination to (i) dynamically select an item subset $Q^*$ that matches a given LM's capability profile—easier items are routed to the weaker LM $m_a$, more difficult items to the stronger LM $m_b$—and (ii) represent a given LM's performance in a latent ability space rather than standard accuracy space. (c, d) Compared to baselines such as evaluating on a random subset of items (Random), Fluid Benchmarking improves benchmarking in various ways: it substantially decreases step-to-step evaluation variance, exemplified by training curves of Pythia-2.8B evaluated on ARC Challenge with 30 items (c), while at the same time increasing the external validity of evaluation, shown as the mean rank distance between an LM's predicted and true rank (d). See text for more details.

et al., 2023; Gupta et al., 2024; Paech, 2024). However, most of these studies have addressed individual aspects of evaluation quality in isolation, sometimes with unintended negative consequences—for example, Madaan et al. (2024) showed that efficient benchmarking methods can increase evaluation variance between training runs with different random seeds, thus reducing benchmarks' practical utility.

In this paper, we propose Fluid Benchmarking, a new benchmarking method that improves evaluation across multiple relevant dimensions. Fluid Benchmarking is based on the insight that the relative value of benchmark items depends on an LM's capability level; for example, a hard question might be too difficult for a weak LM, but informative for a strong LM. Fluid Benchmarking integrates item response theory (IRT; Lord, 1980; van der Linden & Hambleton, 1997; DeMars, 2010), which represents performance in a latent ability space, with methods from computerized adaptive testing used in education (Meijer & Nering, 1999; Chang, 2015; Magis et al., 2017): IRT draws upon existing LM evaluation results to enrich benchmarks with information about item difficulty and discrimination, which is leveraged to dynamically select items that match an LM's capability level (Figure 1). This contrasts with the until now universal practice of what we call *static* benchmarking, which assumes a globally optimal set of evaluation items for all LMs.

In our experiments, we investigate how different methods for improving evaluation affect the *efficiency*, *validity*, *variance*, and *saturation* of benchmarks. We specifically focus on *LM evaluation during pretraining*, a key application of benchmarking. We evaluate six LMs on six benchmarks, comparing Fluid Benchmarking against a broad set of methods proposed in prior work. We find that Fluid Benchmarking consistently outperforms *all* baselines across *all* dimensions of evaluation quality. For example, compared to the common practice of random item sampling, Fluid Benchmarking improves validity and lowers step-to-step variance on MMLU using *fifty times fewer items*. Our analysis attributes these gains to the complementary effects of the two key components of Fluid Benchmarking: IRT enhances validity, while dynamic item selection reduces variance.

## 2 Preliminaries: Benchmark Refinement

In this paper, we introduce **benchmark refinement** as the problem of improving benchmarking by optimizing (i) the selection of evaluation items as well as (ii) the aggregation of their results into benchmark-level scores. We argue that many existing efforts in LM evaluation, previously considered in isolation, can be productively unified under this umbrella.

### 2.1 Evaluation is Selection, Scoring, and Aggregation

Let $m_i$ be an LM that is to be evaluated on a benchmark $Q$. We refer to the elements $q_j \in Q$ as *items*. In a general form, evaluating $m_i$ on $Q$ can be expressed as

$$\text{EVALUATE}(m_i, Q) = \underbrace{\text{AGGREGATE}_{q_j \in \text{SELECT}(Q)}}_{\substack{\text{benchmark-level} \\ \text{aggregation}}} \underbrace{\left(\text{SCORE}(m_i, q_j)\right)}_{\substack{\text{item-level} \\ \text{scoring}}}, \tag{1}$$

where SELECT is a *selection function* that determines the set of evaluation items, SCORE is a *scoring function* that quantifies LM performance on each item in the evaluation set, and AGGREGATE is an *aggregation function* applied over the item-level scores. For notational convenience, we denote the evaluation set as $Q^*$, which may be a subset, superset, or identical to $Q$. If SCORE $\in \{0, 1\}$ is a binary function, AGGREGATE returns the mean of the item-level scores, and SELECT$(Q) = Q$, we recover the standard accuracy metric commonly used in LM benchmarking, which we denote as ACCURACY$(m_i, Q)$.

With Equation 1, we can break down LM evaluation into two components: *item-level scoring* and *benchmark-level aggregation*. While evaluation quality can be improved at both levels, many studies take item-level scores as given and focus on improving benchmark-level aggregation. The present line of work asks: how can we improve LM evaluation through the choice of both (i) the selection function SELECT and/or (ii) the aggregation function AGGREGATE? We refer to this problem as *benchmark refinement*.

### 2.2 Dimensions of Evaluation Quality

What aspects of evaluation can be improved through benchmark refinement? In this paper, we focus on four dimensions, each motivated by prior work:

- **Efficiency.** Evaluation can be made more efficient by selecting a small evaluation set $Q^*$, with $|Q^*| \ll |Q|$. Prior work has explored random sampling (Perlitz et al., 2024a), item clustering (Polo et al., 2024; Vivek et al., 2024), heuristic filtering (Gupta et al., 2024), and information filtering (Kipnis et al., 2025). Several studies have paired this with modifying AGGREGATE (Polo et al., 2024; Kipnis et al., 2025).

- **Validity.** As a means of measuring an *underlying capability* in LMs, a benchmark should be predictive of LM behavior beyond the benchmark itself. Prior work has explored different ways to increase benchmark validity via SELECT—for example, by removing and replacing items in $Q$ that trivially fail to measure the intended capability, such as mislabeled items (Northcutt et al., 2021; Gema et al., 2024; Vendrow et al., 2025).

- **Variance.** If evaluation results on a benchmark fluctuate significantly due to evaluation noise (e.g., metric instability), the benchmark becomes less useful in many practical settings, such as tracking progress during training. While it has been shown that removing items with low discriminative power from $Q$ can reduce variance, attempts to modify AGGREGATE have so far proven less effective (Madaan et al., 2024).

- **Saturation.** Given the rapid improvement in LM capabilities, frontier models often solve most items in benchmarks within a short time, limiting their practical value (Vania et al., 2021; Liang et al., 2023). This saturation has motivated the development of more challenging benchmark variants by choosing SELECT such that it focuses on more difficult items (Suzgun et al., 2023; Gupta et al., 2024; Paech, 2024).

In §4.2, we operationalize each of these dimensions into metrics.

# 3   Methodology: FLUID BENCHMARKING

We introduce FLUID BENCHMARKING, a new method for benchmark refinement that departs from prior work (i) by changing AGGREGATE such that LM performance is represented in a latent *ability space* rather than the standard *accuracy space* (§3.1), and (ii) by choosing SELECT to dynamically adjust the subset of evaluation items to an LM (§3.2).

## 3.1   Measuring Language Model Performance in Latent Ability Space

Over the past several decades, research in psychometrics has developed a suite of methods to address the challenges discussed in the previous section, which arise in a similar form in human testing. We argue that psychometric methods can be fruitfully applied to the evaluation of LMs. In particular, we draw upon item response theory (IRT; Lord, 1980; van der Linden & Hambleton, 1997; DeMars, 2010), which represents test takers in a latent ability space. The specific IRT model we use is a two-parameter logistic (2PL) model (Lord, 1952; Birnbaum, 1968). Before providing a formal definition, we begin with a quick overview of the advantages of IRT-based ability estimates over accuracy.

The key property that distinguishes IRT-based ability estimates from accuracy is that IRT takes *item characteristics* into account, whereas accuracy treats all items equally. In the 2PL model that we consider here, the two item characteristics are:

– **Item difficulty.** Correctly answering an *easy* item has a different impact on ability estimates than correctly answering a *difficult* item.

– **Item discrimination.** Items exhibit varying rates at which the likelihood of a correct response increases with ability. Low-discrimination items are often problematic—for example, we find empirically that many of them are mislabeled (see §6).

These features might be beneficial for benchmark refinement. In terms of *efficiency*, item parameters provide a principled basis for selecting $Q^*$. Item discrimination potentially offers dual benefits: it could enhance *validity* by reducing the impact of mislabeled items, while simultaneously decreasing *variance* by placing less weight on items that inconsistently differentiate between similar LMs. Finally, the fact that difficult items affect ability estimates differently than easy items could delay *saturation* effects, as differences in performance among strong LMs on difficult items are better captured than by accuracy.

**Formulation.** Let $M = \{m_1, \ldots, m_k\}$ be a set of LMs that have been evaluated on a benchmark $Q$. Assuming items with two outcomes, the probability that an LM $m_i$ answers item $q_j$ correctly can be modeled as a Bernoulli random variable $u_{ij}$, where $u_{ij} = 1$ (success) iff the LM's answer is correct. The probability that $u_{ij} = 1$ is modeled as

$$p(u_{ij} = 1) = \text{logistic}\left(a_j(\theta_i - b_j)\right), \tag{2}$$

where the parameter $\theta_i$ corresponds to the ability of LM $m_i$, and the item $q_j$ is characterized by parameters $a_j > 0$ (discrimination) and $b_j$ (difficulty). Equation 2 is commonly visualized using so-called *item characteristic curves* (see Appendix A for examples). For model estimation, we assume local independence and maximize the probability of the full response matrix $U \in \{0, 1\}^{k \times l}$ using Markov chain Monte Carlo (Junker et al., 2016), with hierarchical priors on all parameters as suggested by Natesan et al. (2016).

Given a fitted 2PL model, the item parameters $a_j$ and $b_j$ can be used to estimate the ability $\hat{\theta}_i$ of a previously unevaluated LM $m_i$ by maximizing

$$\hat{\theta}_i = \max_{\theta} \prod_{j=1}^{l} \left[\text{logistic}\left(a_j(\theta - b_j)\right)\right]^{u_{ij}} \left[1 - \text{logistic}\left(a_j(\theta - b_j)\right)\right]^{1 - u_{ij}}. \tag{3}$$

Here, the item parameters $a_j$ and $b_j$ are treated as fixed. We use maximum a posteriori estimation (Birnbaum, 1969) to determine $\hat{\theta}_i$. Equation 3 defines a benchmark-level aggregation as in Equation 1, with $\text{SCORE}(m_i, q_j) = u_{ij}$ and $\text{AGGREGATE}(m_i, Q) = \hat{\theta}_i$. We

denote this form of model evaluation as ABILITY($m_i, Q$), which constitutes one of the two methodological pillars of FLUID BENCHMARKING.

So far we have only modified AGGREGATE, not SELECT, but IRT allows for a principled way to dynamically adapt $Q^*$ to an LM. Next, we present a method how to do so.

## 3.2 Dynamic Selection of Evaluation Items

Benchmarks are used to monitor performance during pretraining, when LMs are undergoing rapid development. Can the same $Q^*$ be optimal for both a near-random word predictor (early in training) and a highly capable model?

One way to approach this question is by examining the informativeness of items with respect to the ability estimate for a given LM, which can be formalized using Fisher information (Reckase, 2009). In the case of the 2PL model, this is given by

$$I(\theta_i, a_j, b_j) = a_j^2 \, \text{logistic} \left( a_j(\theta_i - b_j) \right) \left[ 1 - \text{logistic} \left( a_j(\theta_i - b_j) \right) \right]. \tag{4}$$

It can be shown that items with higher Fisher information yield more precise ability estimates (Reckase, 2009), and they should be prioritized in $Q^*$.

To analyze how the informativeness of items change as a function of LM ability, we examine HellaSwag (Zellers et al., 2019). We consider the scenario of pretraining mentioned above and simulate a training run with 50 checkpoints. In Figure 2, we show how the Fisher information distributes over HellaSwag items as a function of training progress. The subset of items with the highest Fisher information substantially changes over the course of the training run, from very easy items at the beginning of training, to very difficult items at the end of training. These findings suggest that adapting $Q^*$ to the capability level of an LM could result in more precise ability estimates compared to using a static set of items.

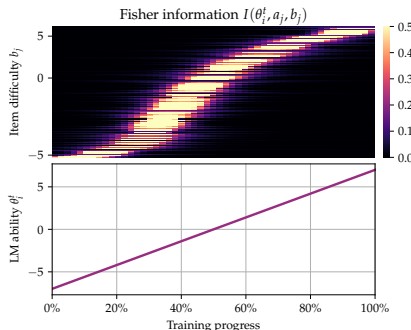

Figure 2: Fisher information (Equation 4) of HellaSwag items as a function of training progress. Lower panel: simulated trajectory of LM ability, which evolves linearly from $\theta_i^1 = -7$ to $\theta_i^{50} = +7$; upper panel: Fisher information of HellaSwag items. The HellaSwag items with highest Fisher information change drastically during training (see Appendix B for more details).

**Formulation.** Inspired by these observations, we draw upon methods developed in the education-research context of computerized adaptive testing (Meijer & Nering, 1999; Chang, 2015; Magis et al., 2017) to adapt $Q^*$ to the capability level of an LM $m_i$. Specifically, we evaluate the LM by iteratively selecting the item from $Q$ with the highest Fisher information given the current ability estimate,

$$Q_i^*(0) = \varnothing; \quad Q_i^*(t) = Q_i^*(t-1) \cup \left\{ \underset{q_j \in Q \setminus Q_i^*(t-1)}{\arg\max} \; I \left( \text{ABILITY}(m_i, Q_i^*(t-1)), a_j, b_j \right) \right\}. \tag{5}$$

We repeat this procedure until the total number of administered items has reached the budgeted size for $Q^*$, at which point we let $Q_i^* = Q_i^*(t)$. Using Equation 5 for SELECT, we compute EVALUATE($m_i, Q$) = ABILITY($m_i, Q_i^*$) as the final evaluation score.

Dynamically selecting items based on Fisher information is expected to reinforce the very properties that make IRT-based methods promising for LM evaluation to begin with. For example, given that $I \propto a_j^2$ (Equation 4), low-discrimination items are unlikely to be included in $Q^*$. Similarly, because $I$ is maximized when $\theta_i = b_j$ (where $I = a_j^2/4$), dynamic selection naturally adapts to the capability level of an LM, evaluating weaker LMs on easier items and stronger LMs on more difficult ones.

## 4 Experiments

### 4.1 Experimental Setup

In this paper, we focus on *LM evaluation during pretraining*. While the four dimensions of benchmark refinement, introduced in §2, are relevant across evaluative settings, pretraining provides a particularly suitable testbed, as it allows for straightforward quantification and measurement of each dimension (see §4.2).

More specifically, we examine the pretraining runs of six LMs with publicly available checkpoints. Our main focus lies on 7B LMs, for which we pick Amber-6.7B (Liu et al., 2023), OLMo1-7B (Groeneveld et al., 2024), OLMo2-7B (OLMo et al., 2025), and Pythia-6.9B (Biderman et al., 2023). We also examine a smaller LM, specifically Pythia-2.8B (Biderman et al., 2023), as well as a larger LM, specifically K2-65B (Liu et al., 2025). For each LM, we evenly select between 61 and 94 checkpoints (see Appendix C for more details).

In terms of benchmarks, we focus on the Open LLM Leaderboard (Beeching et al., 2023), which comprises ARC Challenge (Clark et al., 2018), GSM8K (Cobbe et al., 2021), HellaSwag (Zellers et al., 2019), MMLU (Hendrycks et al., 2021), TruthfulQA (Lin et al., 2022), and Wino-Grande (Sakaguchi et al., 2020). For the IRT models underlying FLUID BENCHMARKING, we fit 2PL models to the evaluation results of LMs contained in the Open LLM Leaderboard. We exclude the six test LMs and related models (e.g., OLMo1-1B), as well as posttrained models, since our experiments focus on evaluation during LM pretraining. This results in a final set of 102 LMs used for IRT model training (see Appendix D for the full inclusion criteria). We fit separate unidimensional IRT models for each benchmark; we initially experimented with multidimensional models as well as a single unidimensional model across all benchmarks, but these approaches yielded worse results (see Appendix E for details).

We then evaluate all checkpoints of the six selected LMs on the six benchmarks and vary the evaluation strategy (see §4.3). In total, we examine 2,802 checkpoint-benchmark combinations, resulting in over 13 million item-level evaluations.

### 4.2 Evaluation Measures

We operationalize the four dimensions of evaluation quality introduced in §2 as follows:

- **Efficiency.** We measure efficiency by systematically varying the number of items used for evaluating on a benchmark (i.e., the size of $Q^*$). We explore a range of subset sizes, varying from 10 to 500 items per benchmark.
- **Validity.** We evaluate validity by testing how well estimated performance on one benchmark predicts performance on a different benchmark that targets the *same* capability. Specifically, we compute the distance between an LMs' predicted ranks on the two benchmarks. We always calculate the rank for the second benchmark based on accuracy. We examine ARC Challenge and MMLU, which assess knowledge and reasoning, and HellaSwag and WinoGrande, which assess commonsense reasoning.
- **Variance.** We measure the step-to-step variance of the training curve for a combination of LM and benchmark. Specifically, let $x_i^t(Q) = \text{EVALUATE}(m_i^t, Q)$ represent the measured performance (e.g., accuracy) on benchmark $Q$ for model $m_i$ at a certain checkpoint $t$. We measure the normalized total variation,

$$\text{TV}(m_i, Q) = \frac{n}{n-1} \times \frac{\sum_{t=1}^{n-1} |x_i^{t+1}(Q) - x_i^t(Q)|}{|x_i^n(Q) - x_i^1(Q)|}, \quad (6)$$

  where a lower value means lower variance and hence better evaluation quality.
- **Saturation.** To measure the saturation of a benchmark under a given evaluation strategy, we compute the monotonicity of the training curve, defined as the absolute Spearman rank correlation between the sequence of checkpoints and the predicted performance values (e.g., accuracies). More monotonic training curves indicate that increased pretraining consistently yields better performance, suggesting that the benchmark has not yet saturated (at least for LMs within the considered capability range).

| Measure | Method | Baseline$_{\text{Items per benchmark}}$ | | | | | |
|---|---|---|---|---|---|---|---|
| | | AP$_{10}$ | AP$_{50}$ | TB$_{100}$ | MB$_{143}$ | SM$_{460}$ | MA$_{1,848}$ |
| Validity | BASELINE | 20.0 | 15.2 | 9.8 | 8.7 | 15.9 | 14.5 |
| *Rank distance* ↓ | FLUID BENCHMARKING | **10.1** | **8.8** | **8.7** | **8.6** | **14.0** | **8.3** |
| Variance | BASELINE | 28.3 | 19.1 | 30.5 | 17.9 | 10.0 | 20.4 |
| *Total variation* ↓ | FLUID BENCHMARKING | **10.7** | **6.5** | **6.1** | **5.5** | **2.8** | **4.8** |
| Saturation | BASELINE | 0.48 | 0.62 | 0.69 | 0.79 | 0.88 | 0.64 |
| *Rank correlation* ↑ | FLUID BENCHMARKING | **0.76** | **0.86** | **0.85** | **0.85** | **0.97** | **0.77** |

Table 1: Comparison against baseline methods. AP: ANCHOR POINTS (Vivek et al., 2024); TB: TINYBENCHMARKS (Polo et al., 2024); MB: METABENCH (Kipnis et al., 2025); SM: SMART (Gupta et al., 2024); MA: MAGI (Paech, 2024). The table shows the results averaged across six benchmarks, six LMs, and between 61 and 94 checkpoints per LM, totaling 2,802 values contributing to each mean. For METABENCH, the number of items is an average across benchmarks, and we exactly match the benchmark-level numbers for the comparison.

### 4.3 Baselines

We compare against several previous benchmark refinement methods. First, we examine ANCHOR POINTS (Vivek et al., 2024), a method for efficient evaluation based on item clustering. We use the Open LLM Leaderboard to cluster the benchmarks and consider two subset sizes in the range examined by the authors (10 and 50). We also examine two IRT-based methods, TINYBENCHMARKS (Polo et al., 2024) and METABENCH (Kipnis et al., 2025), and compare directly against their subsets and evaluation tools. In terms of methods for increasing difficulty, we include the hard versions of ARC Challenge and MMLU from SMART (Gupta et al., 2024) and MAGI (Paech, 2024), respectively.

FLUID BENCHMARKING differs from prior methods through its AGGREGATE (§3.1) and its SELECT (§3.2). To disentangle these factors, we consider a baseline in which we ablate SELECT and compute an ability estimate based on a random subset of items (RANDOM IRT). In addition, we consider a baseline in which we ablate both SELECT and AGGREGATE, using a random subset of items to compute accuracy (RANDOM), a popular approach for efficient evaluation (Liang et al., 2023; Gu et al., 2024; Perlitz et al., 2024a).

## 5 Results

FLUID BENCHMARKING outperforms all baselines across all dimensions and sample sizes, often by a wide margin (see Appendix F for breakdowns by benchmark and LM).

**Validity.** Table 1 (top panel) shows that FLUID BENCHMARKING leads to smaller rank distances than all baselines. It outperforms ANCHOR POINTS, SMART, and MAGI by wide margins, almost halving the mean rank distance of ANCHOR POINTS. The IRT-based methods are better, but FLUID BENCHMARKING still outperforms them.

Table 2 (top panel) shows that ablating the dynamic selection of items (FLUID BENCH-MARKING vs. RANDOM IRT) results in lowered validity, but the gap diminishes with more items. This is expected since (dynamic) $Q_i^*$ approximates (static) $Q^*$ as the number of items increases, resulting in converging ability estimates. Ablating the IRT-based ability estimation (RANDOM vs. RANDOM IRT) leads to a much bigger drop in validity (see Figure 1d), suggesting that the information provided by IRT is particularly beneficial for improving the predictiveness of performance estimates. This is also supported by the high validity of the two IRT-based baselines TINYBENCHMARKS and METABENCH.

**Variance.** Table 1 (mid panel) shows that FLUID BENCHMARKING outperforms all baselines in terms of step-to-step variance. This trend holds consistently across LMs, benchmarks, and subset sizes (see Appendix G for details). Figure 1c illustrates this with the evaluation of Pythia-2.8B on ARC Challenge, using 30 items. Interestingly, the gap between TINY-

| Measure | Method | Items per task | | | |
|---------|--------|------|------|------|------|
| | | 10 | 50 | 100 | 500 |
| Validity | RANDOM | 20.0 | 15.2 | 16.9 | 9.1 |
| *Rank distance ↓* | RANDOM IRT | 14.1 | 11.1 | 10.6 | 8.4 |
| | FLUID BENCHMARKING | **10.1** | **8.8** | **8.7** | **8.3** |
| Variance | RANDOM | 29.0 | 19.1 | 19.8 | 10.2 |
| *Total variation ↓* | RANDOM IRT | 18.2 | 15.7 | 17.8 | 10.9 |
| | FLUID BENCHMARKING | **10.7** | **6.5** | **6.1** | **4.9** |
| Saturation | RANDOM | 0.47 | 0.62 | 0.64 | 0.79 |
| *Rank correlation ↑* | RANDOM IRT | 0.48 | 0.69 | 0.71 | 0.85 |
| | FLUID BENCHMARKING | **0.76** | **0.86** | **0.85** | **0.88** |

Table 2: Comparison against ablated methods. See caption of Table 1 for more details.

BENCHMARKS and METABENCH on the one hand, and the remaining baselines on the other, is much less pronounced than for validity—the two methods even lead to *worse* results for variance (e.g., TINYBENCHMARKS/100 items: 30.5 vs. RANDOM/100 items: 19.8).

Table 2 (mid panel) reflects this trend: the gap is smaller between RANDOM and RANDOM IRT than between RANDOM IRT and FLUID BENCHMARKING—on 500 items, RANDOM IRT even leads to a *higher* variance than RANDOM. This suggests that the key to FLUID BENCHMARKING's low variance lies in its dynamic item selection, which is consistent with psychometric theory: since the variance of ability estimates is inversely proportional to test information (Lord, 1983), and since FLUID BENCHMARKING selects highly informative items, the resulting measurement error is substantially reduced.

**Saturation.** Tables 1 and 2 show that FLUID BENCHMARKING consistently outperforms all baselines in terms of saturation as well (see Appendix G for details). SMART and MAGI perform better than some of the other baselines, suggesting that these methods partially mitigate the saturation problem, yet FLUID BENCHMARKING addresses it more effectively.

**Efficiency.** Taking a global look at the results, we observe that FLUID BENCHMARKING leads to improvements across all subset sizes, but is especially effective for small sample sizes. For example, with 500 items FLUID BENCHMARKING improves the mean rank distance (validity) of RANDOM by 0.8, but with 10 items by 9.9.

In Appendix H, we show that FLUID BENCHMARKING can improve evaluation quality even when efficiency is not a concern, outperforming full-benchmark accuracy.

## 6 Analysis and Discussion

**FLUID BENCHMARKING Avoids Mislabeled Items.** To test whether FLUID BENCHMARKING indeed avoids problematic instances such as mislabeled questions, we leverage MMLU-Redux (Gema et al., 2024), a recent effort that annotated MMLU questions for label errors. We compute the average number of mislabeled items in FLUID BENCHMARKING and RANDOM ($|Q^*| = 100$) across all LMs and checkpoints, finding that it is *nearly two orders of magnitude smaller* in the former (0.01) than in the latter (0.75)—in other words, while it takes roughly 100 benchmarking sessions for a mislabeled item to appear with FLUID BENCHMARKING, one occurs in nearly every session with RANDOM. This suggests that FLUID BENCHMARKING is highly effective at avoiding mislabeled items.

**FLUID BENCHMARKING Adapts Items to Language Model Capability.** To test whether item selection indeed dynamically adapts to the capability level of a given LM, we analyze how item selection changes as an LM gets better over the course of pretraining. Figure 3 visualizes the items selected for FLUID BENCHMARKING ($|Q^*| = 50$) with OLMo1-7B evaluated on HellaSwag. We observe a substantial shift in the selected items: initially, items are very easy, but they get gradually more difficult as the LM improves.

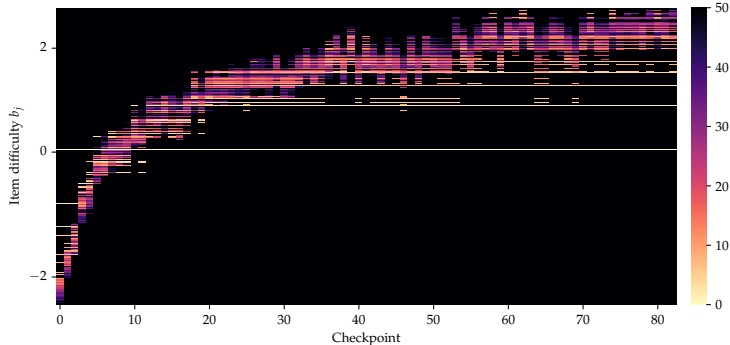

Figure 3: FLUID BENCHMARKING of OLMo1-7B (HellaSwag/50 items). The figure shows items (stacked along $y$-axis) selected for FLUID BENCHMARKING as a function of different checkpoints. Items are ordered by difficulty $b_j$. Items selected for FLUID BENCHMARKING are colored by time of selection; brighter colors reflect earlier appearance during evaluation. The bright line close to $y = 0$ represents the first item, which is always the same. Depending on how the LM responds, the next item is either easier (incorrect response, see first few checkpoints) or more difficult (correct response, see checkpoints after 11).

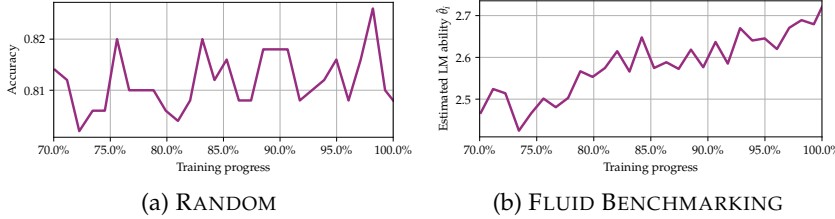

(a) RANDOM        (b) FLUID BENCHMARKING

Figure 4: Training curves of OLMo2-7B (HellaSwag/500 items) with RANDOM (a) and FLUID BENCHMARKING (b). The figures plot the final 30% of training. While performance in accuracy space shows no meaningful improvement (a), performance in ability space continues to provide a clear learning signal through the end of training (b).

**FLUID BENCHMARKING Delays Onset of Benchmark Saturation.** To test whether FLUID BENCHMARKING indeed delays the onset of benchmark saturation, we focus on HellaSwag ($|Q^*| = 500$). Figure 4a shows OLMo2-7B's performance during the final 30% of the training run, measured with RANDOM. Performance is already high by the 70% mark and does not show a consistent upward trend thereafter, instead fluctuating around the same level. By contrast, with FLUID BENCHMARKING (see Figure 4b), performance continues to improve steadily through the end of training, suggesting that FLUID BENCHMARKING effectively mitigates early benchmark saturation. This difference is captured by our measure of saturation: for the entire training run, the monotonicity of the HellaSwag curve is 0.91 for RANDOM, compared to 0.99 for FLUID BENCHMARKING.

**Dynamic Stopping.** A further advantage of FLUID BENCHMARKING is its support for dynamic stopping. In Figure 5, we demonstrate this with OLMo1-7B and HellaSwag, where we use the standard error of the ability estimate as the stopping criterion (Magis et al., 2017). Specifically, we terminate the evaluation once the standard error falls below the average ability gap between two rank-adjacent LMs on the Open LLM Leaderboard. The number of items required to reach this precision varies substantially over training, from around 20 at the beginning to over 80 midway, indicating that the common practice of using a fixed number of evaluation items is suboptimal.

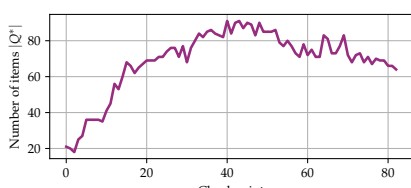

Figure 5: FLUID BENCHMARKING with dynamic stopping on OLMo1-7B/HellaSwag (see text for details).

**The False False Promise of Item Response Theory.** Madaan et al. (2024) criticized IRT-based benchmark refinement methods for increasing variance, speaking of a "false promise of item response theory" for LMs. Our findings contextualize this in crucial ways. On the one hand, we confirm Madaan et al. (2024)'s observation that IRT-based methods (Polo et al., 2024; Kipnis et al., 2025) increase step-to-step variance. On the other hand, our results demonstrate that the issue is not intrinsic to IRT itself, but rather arises from the fact that prior IRT-based methods have not fully leveraged a central strength of IRT: dynamically adapting items to the LM's capability. We find that exploiting this potential substantially reduces variance compared to accuracy-based evaluations.

**Extension to Other Settings.** While we focus on LM evaluation during pretraining in this paper, where efficiency is especially critical due to high computational costs and the need for frequent in-loop evaluations, FLUID BENCHMARKING is not inherently limited to this phase and holds potential value for posttraining as well. Furthermore, FLUID BENCHMARKING is readily extendable to other languages and modalities, provided that evaluation results are available to fit an IRT model. For example, applying FLUID BENCHMARKING to vision-language models could leverage leaderboards such as VHELM (Lee et al., 2024).

**Generalization Beyond Train Language Models.** While IRT ability estimates are not inherently upper bounded by the abilities of the train LMs (i.e., the LMs used to estimate item parameters), the utility of FLUID BENCHMARKING still depends on having stable and up-to-date IRT models, especially given the rapid pace of LM development. Consider the subset of benchmark items that were not answered correctly by any train LM. These items are effectively assigned the same maximum difficulty. If we conduct FLUID BENCHMARKING with a new LM that is better than any train LM, evaluation will quickly move to those most difficult items. However, a fixed IRT model cannot distinguish finer levels of difficulty among them. Therefore, IRT models used for FLUID BENCHMARKING should be regularly updated with fresh evidence. We hope that the IRT models released as part of this paper can serve as a starting point for such an extensible reference standard.

## 7 Related Work

Our study adds to the growing body of work on **benchmark refinement** (see §2 for details). Besides providing a formal definition of this emerging field, we introduce a method that improves benchmarking across multiple dimensions.

Prior work has used **IRT models in natural language processing** (Lalor et al., 2016; 2018; 2019; Lalor & Yu, 2020; Rodriguez et al., 2021; Vania et al., 2021; Rodriguez et al., 2022; Lalor et al., 2024). Recently, there have been several attempt to use IRT in the context of benchmark refinement, to improve efficiency (Polo et al., 2024; Kipnis et al., 2025) and mitigate benchmark saturation (Paech, 2024). Our work differs by considering a wider set of criteria and focusing on evaluation during pretraining; we also show that static benchmarks forego the full potential of IRT, which lies in the possibility of adaptive testing.

So far, uses of **adaptive testing in natural language processing** have been confined to improving the cold start problem (Rodriguez et al., 2021).

## 8 Conclusion

In this work, we unify disparate lines of research to introduce the general problem of *benchmark refinement*. We define four key dimensions along which benchmark refinement methods should be evaluated: efficiency, validity, variance, and saturation. We introduce FLUID BENCHMARKING, a new benchmarking method that combines item response theory with adaptive testing, improving over prior approaches along all dimensions. In a recent perspective, Zhuang et al. (2024) argued that adaptive testing "will become the new norm in AI model evaluation," but so far a large-scale analysis of its potential as a general evaluation method has been missing. Our study is the first to provide this analysis and establishes a foundation for new, exciting research in AI evaluation methodology.

## Acknowledgments

This material is based upon work supported by the U.S. National Science Foundation (#2113530, #2313998). Any expressed opinions, findings, and conclusions or recommendations are those of the author(s) and do not necessarily reflect the views of the U.S. National Science Foundation. IM was supported by the NSF CSGrad4US Fellowship. PWK was supported by the Singapore National Research Foundation and the National AI Group in the Singapore Ministry of Digital Development and Information under the AI Visiting Professorship Programme (#AIVP-2024-001) and the AI2050 program at Schmidt Sciences. Our special thanks go to the members of AllenNLP, Oyvind Tafjord, and Sarah Wiegreffe for insightful discussions, as well as to the reviewers for their valuable feedback.

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

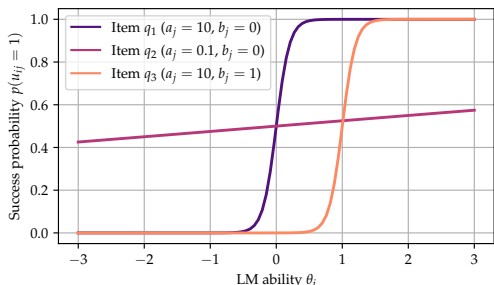

Figure 6: Example item characteristic curves. The $x$-axis shows the ability parameter $\theta_i$; the greater $\theta_i$, the higher the success probability $p(u_{ij} = 1)$. The difficulty parameter $b_j$ indicates the value of $\theta_i$ at which $p(u_{ij} = 1) = 0.5$, reflected by the location of the curve (compare $q_1$ vs. $q_3$). The discrimination parameters indicates how sharply $p(u_{ij} = 1)$ changes when $\theta_i$ is close to $b_j$. $a_j$ is proportional to the slope of the curve (compare $q_1$ vs. $q_2$). When the curve is flat (i.e., low $a_j$), this implies that even some high-ability LMs failed on this item.

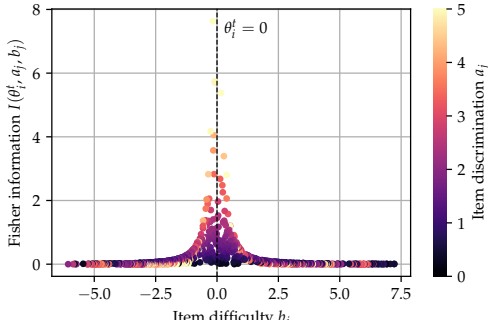

Figure 7: Fisher information of HellaSwag items halfway through the simulated training run, when $\theta_i^t = 0$. The figure corresponds to the distribution obtained by taking a vertical slice through Figure 2 at $\theta_i^t = 0$. In line with Equation 4, Fisher information is highest for items whose difficulty $b_j$ is close to $\theta_i^t$. It also increases with item discrimination $a_j$, an effect that is particularly pronounced when $b_j \approx \theta_i^t$. By contrast, when $b_j$ is far from $\theta_i^t$, higher discrimination has an only modest effect on Fisher information.

## A  Item Characteristic Curves

We provide example item characteristic curves in Figure 6.

## B  Fisher Information of HellaSwag Items

For illustrative purposes, Figure 7 shows the Fisher information of HellaSwag items halfway through the simulated training run, when $\theta_i^t = 0$.

## C  Checkpoint Details

We provide details about the selected LM checkpoints. For Amber-6.7B, we select 73 checkpoints. For OLMo1-7B, we select 83 checkpoints. For OLMo2-7B, we select 94 checkpoints. For Pythia-6.9B, we select 78 checkpoints. For Pythia-2.8B, we select 78 checkpoints. For K2-65B, we select 61 checkpoints. For all LMs, checkpoints are selected to ensure even coverage throughout the entire training run.

| | | Benchmark | | | | | |
|---|---|---|---|---|---|---|---|
| Measure | Method | ARC | GSM | HS | MMLU | TQA | WG |
| Validity | RANDOM | 21.9 | — | 12.9 | 20.5 | — | 12.4 |
| *Rank distance* ↓ | RANDOM IRT | 15.9 | — | 5.0 | 13.4 | — | 8.2 |
| | FLUID BENCHMARKING | **14.5** | — | **4.5** | **10.7** | — | **4.9** |
| Variance | RANDOM | 10.2 | 22.2 | 3.8 | 49.7 | 18.1 | 14.6 |
| *Total variation* ↓ | RANDOM IRT | 7.9 | 28.9 | 12.0 | 20.8 | 15.1 | 22.1 |
| | FLUID BENCHMARKING | **3.3** | **9.1** | **2.0** | **6.3** | **9.8** | **5.8** |
| Saturation | RANDOM | 0.75 | 0.66 | 0.88 | 0.51 | 0.43 | 0.61 |
| *Rank correlation* ↑ | RANDOM IRT | 0.82 | 0.60 | 0.88 | 0.56 | 0.63 | 0.76 |
| | FLUID BENCHMARKING | **0.95** | **0.86** | **0.98** | **0.67** | **0.71** | **0.93** |

Table 3: Comparison against baselines, split by benchmark. ARC: ARC Challenge; GSM: GSM8K; HS: HellaSwag; TQA: TruthfulQA; WG: WinoGrande.

## D Language Model Inclusion Criteria

We used the following criteria when selecting LMs for IRT model training:

– We only included pretrained LMs. Finetuned, merged, fused, distilled, or continually pretrained LMs were excluded, as they can lead to clusters of highly similar models, potentially skewing the IRT model.

– In the rare cases where an LM appears on the Open LLM Leaderboard with multiple checkpoints, we used only the final checkpoint listed.

– We excluded LMs trained solely on non-English data, but multilingual LMs were included as long as English data were part of their training corpus.

– We removed any LMs from the same model family as the test LMs (e.g., OLMo1-1B).

## E Item Response Model Details

In the main experiments, we fit *separate unidimensional* IRT models *to each benchmark*. Initially, we also experimented with two alternative setups:

– We experimented with fitting a *single unidimensional* IRT model *across all benchmarks*, following prior work suggesting that one latent trait can capture overall model behavior (Kipnis et al., 2025). However, we found that this substantially reduced construct validity. For example, the performance of Amber-6.7B on TruthfulQA decreases during pretraining (Liu et al., 2023); by contrast, when we evaluated Amber-6.7B using a unidimensional IRT model trained across all benchmarks, the estimated ability increased—the IRT model effectively emphasized TruthfulQA items aligned with general trends, obscuring the fact that Amber-6.7B actually becomes *less* truthful during pretraining.

– We experimented with fitting *separate multidimensional* IRT models (with two to five latent traits) *to each benchmark*. These models, however, did not yield consistent improvements in model fit compared to the unidimensional IRT models.

Ultimately, fitting separate unidimensional IRT models to each benchmark offered the best trade-off in our experiments. That said, multidimensional IRT models may offer greater advantages in other settings (e.g., when evaluating multimodal models).

## F Breakdown of Results by Benchmark and Language Model

Table 3 breaks the comparison against baselines down by benchmark. Table 4 breaks the comparison against baselines down by LM. We examine the ablated baselines here, fixing the number of items per benchmark to 100.

| Measure | Method | Language model | | | | | |
|---------|--------|------|-------|-------|-------|------|------|
| | | A-7B | K-65B | O1-7B | O2-7B | P-3B | P-7B |
| Validity | RANDOM | 25.5 | 5.1 | 10.7 | 7.1 | 23.1 | 28.1 |
| *Rank distance* ↓ | RANDOM IRT | 20.2 | 5.2 | 8.0 | 6.1 | 8.3 | 15.3 |
| | FLUID BENCHMARKING | **19.3** | **2.1** | **6.7** | **3.4** | **8.1** | **11.6** |
| Variance | RANDOM | 21.8 | 14.7 | 10.2 | 15.4 | 35.7 | 20.9 |
| *Total variation* ↓ | RANDOM IRT | 16.2 | 27.0 | 11.4 | 12.4 | 26.1 | 13.7 |
| | FLUID BENCHMARKING | **5.5** | **7.1** | **5.8** | **6.8** | **6.5** | **4.5** |
| Saturation | RANDOM | 0.47 | 0.65 | 0.77 | 0.63 | 0.62 | 0.71 |
| *Rank correlation* ↑ | RANDOM IRT | 0.66 | 0.73 | 0.83 | 0.63 | 0.67 | 0.73 |
| | FLUID BENCHMARKING | **0.82** | **0.89** | **0.91** | **0.80** | **0.81** | **0.87** |

Table 4: Comparison against baselines, split by LM. A-7B: Amber-7B; K-65B: K2-65B; O1-7B: OLMo1-7B; O2-7B: OLMo2-7B; P-3B: Pythia-2.8B; P-7B: Pythia-6.9B.

## G   Variance and Saturation Plots

Figure 8 provides a more detailed comparison of FLUID BENCHMARKING and RANDOM in terms of variance (Figure 8a) and saturation (Figure 8b). FLUID BENCHMARKING improves on RANDOM for almost all combinations of benchmark, subset size, and LM.

## H   Comparison Against Full-Benchmark Accuracy

We have shown that FLUID BENCHMARKING improves evaluation quality in terms of validity, variance, and saturation, compared against alternative evaluation methods using the same number of items. Do these advantages persist when evaluation cost is not a concern (i.e., when it is feasible to evaluate on the full set of benchmark items)? To test this, we compare FLUID BENCHMARKING ($|Q^*| = 500$) with full-benchmark accuracy, using the same LMs and benchmarks as in our main experiments (see §4).

We find that full-benchmark accuracy performs worse than FLUID BENCHMARKING across all three evaluation dimensions, despite using substantially more items. This holds for validity (9.1 vs. 8.3 for FLUID BENCHMARKING), variance (23.8 vs. 4.9 for FLUID BENCHMARKING), and saturation (0.85 vs. 0.88 for FLUID BENCHMARKING). Notably, even FLUID BENCHMARKING with only 50 items outperforms full-benchmark accuracy on all three dimensions (cf. Table 2). These results suggest that FLUID BENCHMARKING can improve evaluation quality even in settings where efficiency is not a limiting factor.

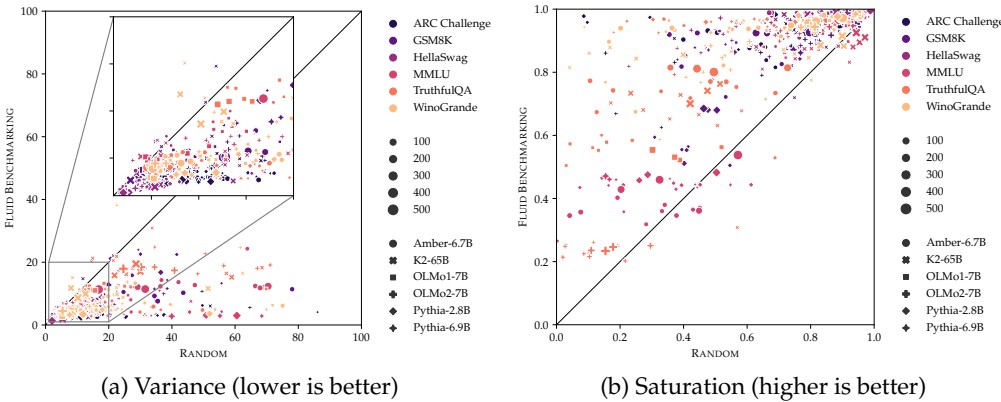

(a) Variance (lower is better)          (b) Saturation (higher is better)

Figure 8: Variance and saturation results. The figure shows pairwise comparisons measuring the total variation (a) and monotonicity (b) of training curves based on RANDOM and FLUID BENCHMARKING. For variance, lower total variation is better. For saturation, high monotonicity is better, as it indicates that increased pretraining consistently yields better performance, suggesting that the benchmark has not yet saturated. Thus, for variance, points in the lower right triangle indicate that FLUID BENCHMARKING is better than RANDOM, and for saturation, points in the upper left triangle indicate that FLUID BENCHMARKING is better than RANDOM. FLUID BENCHMARKING improves on RANDOM for almost all combinations of benchmark, subset size, and LM.

