# OpenReview forum: "Fluid Language Model Benchmarking"
_colmweb.org/COLM/2025/Conference — COLM 2025_

### Official Review · Reviewer_22Eb · 2025-05-13

**Rating:** 7
**Confidence:** 4
**Ethics Flag:** 1

**Summary:**

The central idea in this paper is to use item response theory (IRT) and estimate model capability and item difficulty/discrimination at the same time, all of which are latent variables within the IRT model. The estimated latent capability is then used as the number representing model's performance rather than accuracy aggregated over correct/incorrect responses on a benchmark. The author also show that this method can be used to dynamically select items from a benchmark for evaluation, with applications to 1) track/visualize the model's performance over pre-training, 2) reduce the number of items to estimate a model's capability, and 3) avoid "bad" or not-so-informative items, such as mislabeled ones, in a benchmark.

The way it works is essentially 1) select a number of models, 2) select a benchmark, 3) run each model on this benchmark to get its binary response to each item, and 4) fit an IRT model on this response matrix. For dynamic item selection, the method works by selecting the most "informative" item at each step that maximizes the fisher information between the item and the current estimation of the model's capability.

**Questions To Authors:**

1. What are the benefits of benchmark refinement if cost is not a concern (compared to the cost of pre-training) and we can easily afford to run the evaluation on all items to estimate the model's capability?

2. What happens if you are allowed to run each model on all items of a benchmark? Do the proposed method and the baselines converge? How does the proposed method compares to if you were to represent the model's capability/rank in terms of accuracy, when running the evaluation on all items of a benchmark?

3. What happens if you use the already estimated item parameters to benchmark much more powerful models, such as the latest models such as R1, Qwen 3, Llama 4, and perhaps closed-source ones? How does the capability estimation from the latent IRT parameter compared to if you were to use accuracy aggregated over all items on a given benchmark: e.g. do they correlate well?

**Reasons To Accept:**

This is a really nice application of a very well known concept in education to LLM benchmarking. The idea itself is intuitive and works well, with substantive empirical evidence to back it. Also nicely written and easy to follow. Overall a good paper.

**Reasons To Reject:**

1. From my understanding, the cost of benchmarking is trivial compared to the cost of pre-training. It is not super clear to me what the benefits of benchmark refinement are.

2. This work assumes the item parameters are fixed after learning. These parameters are estimated based on the capabilities of the "students" solving these problems, where each "student" is a model in this case and the problems are items in a benchmark. Therefore, my understanding (I could be wrong) is that the estimated model capability is upper bounded by the best model used to estimate these parameters. Therefore, if a better model is being evaluated, whose "ability" is "better" than all the models used to compute the item parameters, then we may not get a very good estimate of this model's capability. This is concerning because 1) the models used to compute the item parameters are small and not representative of the very best in the open-source community, and 2) models are always improving. As a result, when using this method to benchmark models that are better than those used to estimate IRT parameters, the estimation may not be very accurate.

---

> ### Author Response · Authors · 2025-06-03
> **Response to Reviewer 22Eb**
>
> Thank you for your constructive and encouraging review -- we appreciate your comments and the time you took to evaluate our work!
>
> # Benefits of benchmark refinement
>
> Thank you for raising this matter.
>
> First, while a single evaluation run may appear inexpensive relative to pretraining, evaluation becomes significantly more costly when it is repeated frequently, as is typical in modern LM workflows. For example, evaluating a single model across the full HELM benchmark suite can cost over 4,000 GPU hours or $10,000 in API usage ([Liang et al., 2023](https://openreview.net/forum?id=iO4LZibEqW)). Costs escalate further in in-loop setups, where frequent evaluations are needed during pretraining. In fact, when pretraining a 1B model and evaluating every 100 steps using the OLMES evaluation framework ([Gu et al., 2025](https://aclanthology.org/2025.findings-naacl.282/)), the evaluation time is longer than the pretraining time.
>
> Second, benchmark refinement is not only about _efficiency_, but also about improving other aspects, particularly _validity_, _variance_, and _saturation_ -- critical properties for meaningful model evaluation. Gains in these dimensions can be orthogonal to efficiency: for example, even when evaluating on _all_ items of a benchmark, Fluid Benchmarking puts less weight on mislabeled items, leading to a higher validity empirically (see the next point below).
>
> We appreciate the opportunity to clarify and will include a more explicit discussion of these points in the final version of the paper.
>
> # Full-benchmark evaluation
>
> This is an excellent question -- thank you for raising it.
>
> When evaluation is performed on all items of a benchmark, the dynamic item selection of Fluid Benchmarking is neutralized. In this scenario, our method effectively reduces to the Random IRT baseline, which applies IRT-based ability estimation across all items. Crucially, even in such a full evaluation regime, IRT-based ability estimation offers significant advantages over raw accuracy. Specifically, while accuracy treats all items equally, IRT takes into account item difficulty and assigns less weight to mislabeled items.
>
> Empirically, this distinction is reflected by consistent gains across validity, variance, and saturation when comparing IRT-based ability estimation vs. standard accuracy on all items:
>
> | Evaluation measure   | Accuracy | Ability (IRT) |
> |-------------|-------------|-----------------------|
> | Validity (rank distance; lower is better)   | 9.1              | 7.9                 |
> | Variance (total variation; lower is better)  | 10.0              | 8.7                  |
> | Saturation (rank correlation; higher is better) | 0.85              | 0.88                 |
>
> Comparing these numbers with Fluid Benchmarking (Table 2 in the paper), we observe that Fluid Benchmarking achieves similar (validity, saturation) or even better results (variance) than the full-benchmark IRT ability estimate, with much fewer items. The advantage becomes even more striking when comparing Fluid Benchmarking to full-benchmark accuracy: with just 50 items, Fluid Benchmarking outperforms full-benchmark accuracy across all three evaluation dimensions: validity (8.8 vs. 9.1), variance (6.5 vs. 10.0), and saturation (0.86 vs. 0.85). In other words, even if efficiency is not a concern, Fluid Benchmarking can improve evaluation quality.
>
> We agree that these results provide a valuable contextualization, and we will include them in the final version of the paper.
>
> # Generalization beyond train models
>
> Please note that IRT ability estimates are _not upper bounded_ by the abilities of the train models (i.e., the models used to estimate item parameters). For instance, if the most capable train model correctly answered $k$ items from a benchmark, a new model that correctly answers those $k$ plus at least one additional item will receive a higher ability estimate.
>
> That being said, you are absolutely right that fixed IRT models can be limited in how they adapt to better models. Specifically, consider the subset of items of a benchmark that were not answered correctly by any train model. These items are effectively assigned the same maximum difficulty. If we conduct Fluid Benchmarking with a model that is better than any train model, evaluation will quickly move to those most difficult items. However, the fixed IRT model cannot distinguish finer levels of difficulty among them. Therefore, in this extreme region, ability estimation reduces to something close to accuracy over those hardest items, rather than leveraging graded item properties to make finer distinctions.
>
> This does not invalidate the IRT estimates and hence is not a terrible mode of failure, we believe, but it should be clearly understood by users of the method. Future work could explore extensions such as dynamic recalibration of item parameters. We will include a discussion in the final version of the paper.

---

> ### Comment · Reviewer_22Eb · 2025-06-10
> **Thanks for the response**
>
> I have no further questions and will keep my recommendation to accept.

---

### Official Review · Reviewer_5T1x · 2025-05-15

**Rating:** 9
**Confidence:** 4
**Ethics Flag:** 1

**Summary:**

* The authors propose Fluid Benchmarking, a new LLM evaluation framework inspired by Item Response Theory (IRT).
* The method combines a two-parameter logistic (2PL) IRT model—capturing item difficulty and discrimination—with a novel AGGREGATE function (latent ability estimation) and a SELECTION strategy (adaptive item selection) to improve benchmarking.
* They demonstrate that Fluid Benchmarking outperforms both traditional random sampling and existing IRT-based methods across four key dimensions: efficiency, validity, variance, and saturation.
* Notably, dynamic selection improves precision and reduces variance, while IRT-based scoring improves validity. For example, the approach can detect mislabeled via item-level discremenation.

**Questions To Authors:**

* Figure 2 (upper panel): While it’s clear that the figure illustrates how Fisher information varies over training progress, the distinction between Fisher information, item discrimination, and their roles in item selection may be confusing at first glance. A brief clarification in the caption or main text would help readers unfamiliar with IRT.
* Dynamic stopping criterion: Currently, the number of evaluation items (i.e., benchmark size) must be specified ahead of time. It would be valuable to explore adaptive stopping rules—for example, stopping when the ability estimate converges or when additional items provide diminishing information gain.
* Multi-trait benchmarks: Many real-world benchmarks implicitly test multiple capabilities (e.g., reasoning, commonsense, factual recall). Can Fluid Benchmarking be extended to multi-dimensional IRT (MIRT) models to disentangle and score these traits separately? Alternatively, could single benchmarks be decomposed into interpretable skill dimensions?

**Reasons To Accept:**

* The proposed Fluid Benchmarking framework is grounded in a well-established theoretical foundation—Item Response Theory (IRT), specifically the 2-parameter logistic (2PL) model, which has decades of use in psychometrics.
* It addresses several critical challenges in modern LLM benchmarking, including efficiency, validity, variance, and saturation, which are often treated separately in prior work.
* Empirically, the framework outperforms both traditional baselines and recent IRT-based methods across all four dimensions on multiple benchmarks and model families.

**Reasons To Reject:**

* The current scope of Fluid Benchmarking is limited to pretraining evaluation. While pretraining is a natural setting for measuring dynamic changes in model ability and efficiency is important, the method itself is not inherently restricted to this phase. Extending the framework to post-training settings (e.g., instruction tuning, domain adaptation, or RLHF) would broaden its applicability and strengthen its practical impact.
* Since Fluid Benchmarking relies on latent ability estimation using IRT, the quality and stability of its outputs depend heavily on the LLMs and benchmarks used to fit the IRT model. Although the Open LLM Leaderboard is a good choice, the rapidly evolving LLM landscape means that item parameters and ability scores may become outdated quickly. To ensure wider adoption and consistency, the community would benefit from a standardized and regularly updated IRT reference model.

---

> ### Author Response · Authors · 2025-06-03
> **Response to Reviewer 5T1x**
>
> Thank you for your kind and thoughtful review of our paper -- we really appreciate the positive feedback and helpful suggestions!
>
> # Fluid Benchmarking for post-training
>
> We fully agree that Fluid Benchmarking is not inherently limited to pretraining and has potential value in post-training settings as well. We focused on pretraining here because evaluation efficiency is especially critical in this phase, due to both the high cost of runs and the central role of repeated in-loop evaluation. That being said, we see extending the framework to post-training as an exciting next step and will add a discussion in the final version of the paper.
>
> # Standardized IRT reference models
>
> Thank you for raising this important point. We agree that the utility of Fluid Benchmarking depends on stable and up-to-date IRT models. To support wider adoption, we will release all IRT models trained for this paper, which we hope can serve as a starting point for a standardized reference. We will place more emphasis on our released artifacts in the final version of the paper, and we will also emphasize the importance of regularly building new IRT models on fresh evidence.
>
> # Clarification for Figure 2
>
> Thanks for pointing this out. We will add a clarification to the caption of Figure 2. Additionally, we will include a supplementary figure in the appendix that illustrates how item discrimination influences Fisher information for a given ability level.
>
> # Dynamic stopping criterion
>
> This is an excellent suggestion. We did in fact experiment with dynamically stopping the evaluation once the standard error of the provisional ability estimate dropped below a predefined threshold value. We found that the number of items required to meet this criterion varied significantly over the course of pretraining: earlier checkpoints required fewer items, while later ones needed more. For example, on HellaSwag, using a maximum standard error of 0.1, only 16 items were needed at the first checkpoint, compared to 32 at the final checkpoint. We will include these results in the final version of the paper.
>
> # Multidimensional IRT models
>
> Thank you for raising this. We indeed experimented with fitting separate multidimensional IRT (MIRT) models with two to five latent traits to each benchmark to investigate whether they could capture multiple underlying capabilities. However, the MIRT models did not yield improvements in model fit over unidimensional IRT models, which led us to not pursue this direction further. We will report these results in the appendix. In general, we fully agree with you that a systematic exploration of dimensionality in LM-based IRT models is a promising avenue for future work, and we will add a discussion in the final version of the paper.

---

### Official Review · Reviewer_JcBQ · 2025-05-16

**Rating:** 7
**Confidence:** 3
**Ethics Flag:** 1

**Summary:**

FLUID BENCHMARKING​ introduces a novel approach to evaluating language models (LMs) by combining ​Item Response Theory (IRT)​​ and ​dynamic item selection​ to address key challenges in LM benchmarking:

​High costs​ of comprehensive evaluations.
​Poor validity​ due to misaligned benchmarks and labeling errors.
​High variance​ in evaluation results.
​Benchmark saturation​ (models scoring near maximum).

The method dynamically selects evaluation items based on an LM’s capability level, improving efficiency (fewer items needed), validity (better rank prediction across benchmarks), and stability (lower variance), while delaying saturation. Experiments on six benchmarks (e.g., MMLU, HellaSwag) and six LMs (e.g., Pythia, OLMo) show consistent improvements over static and IRT-based baselines.

**Questions To Authors:**

- How does FLUID BENCHMARKING guard against inadvertently overfitting to the "known" capability trajectory of LMs during pretraining? Could dynamic item selection bias the evaluation toward reinforcing existing LM weaknesses/strengths rather than holistically measuring progress?
- Traditional IRT assumes unidimensional ability (a single latent skill). How valid is this assumption for multimodal LMs or models with heterogeneous capabilities (e.g., strong in reasoning but weak in factual recall)? Could multidimensional IRT better capture LM capabilities?
- FLUID BENCHMARKING relies on historical LM evaluation data to estimate item parameters. How would the method perform for evaluating novel LM architectures or tasks with no prior data? Is there a risk of perpetuating bias toward existing paradigms?

**Reasons To Accept:**

- Addresses multiple benchmarking issues (efficiency, validity, variance, saturation) in a unified framework.
- Outperforms baselines across all metrics
- Leverages well-established IRT and adaptive testing principles, ensuring methodological rigor.

**Reasons To Reject:**

- Effectiveness on non-English or multimodal benchmarks remains unverified.

---

> ### Author Response · Authors · 2025-06-03
> **Response to Reviewer JcBQ**
>
> Thank you for your thorough review of our paper -- we very much appreciate the positive and constructive comments!
>
> # Non-English and multimodal benchmarks
>
> Thank you for suggesting to evaluate our framework on other languages and modalities. Extending Fluid Benchmarking to these settings is indeed feasible, provided that evaluation results to fit an IRT model are available. For example, Fluid Benchmarking of vision-language models could leverage leaderboards like [VHELM](https://crfm.stanford.edu/helm/vhelm/latest/). While a comprehensive evaluation is beyond the scope of this paper, we will include a discussion in the final version to clarify how our method could be applied more broadly, including potential challenges.
>
> # Overfitting to known capabilities and capability trajectories
>
> Thank you for raising this. Fluid Benchmarking, and IRT-based evaluations more generally, rely on historical evaluation data, which implicitly assumes that future models will exhibit broadly similar capability patterns -- for example, not being dramatically better at the hardest items while worse at the easiest ones. While we have not observed such shifts empirically, they are theoretically possible. In such cases, retraining the IRT models would be necessary to maintain evaluation accuracy. We will add a discussion and recommend ways to mitigate this risk.
>
> (Please note that the pretraining trajectories are not used -- neither during IRT parameter estimation nor in dynamic item selection -- so they cannot directly bias the evaluation.)
>
> # Validity of unidimensional IRT model
>
> Thank you for raising this important point. We considered the question of dimensionality at several stages of the project.
>
> We first experimented with fitting a _single unidimensional_ IRT model across _all_ benchmarks, following prior work suggesting that one latent trait can capture overall model behavior ([Kipnis et al., 2025](https://openreview.net/forum?id=4T33izzFpK)). However, we found that this substantially reduced construct validity. For example, the performance of Amber-6.7B on TruthfulQA decreases during pretraining ([Liu et al., 2023](https://arxiv.org/abs/2312.06550)). By contrast, when we evaluated Amber-6.7B using a unidimensional IRT model trained across all benchmarks, the estimated ability increased -- the IRT model effectively emphasized TruthfulQA items aligned with general trends, obscuring the fact that Amber-6.7B actually becomes _less_ truthful during pretraining.
>
> Next, we fit _separate multidimensional_ IRT models (with two to five latent traits) to _each_ benchmark. These models, however, did not yield consistent improvements in model fit compared to unidimensional IRT models.
>
> Ultimately, we adopted a compromise: fitting _separate unidimensional_ IRT models to _each_ benchmark. This struck the best balance between interpretability and model fit.
>
> We realize now that we should make this line of analysis more explicit, and we will add the additional experiments in the appendix. In general, we fully agree with you that a systematic exploration of dimensionality in LM-based IRT models is a promising avenue for future work, especially in settings where heterogeneous capabilities are expected to play a central role. We will incorporate a discussion of this much-needed future work in the final version of the paper.
>
> # Novel architectures and tasks with no prior data
>
> These are excellent points.
>
> Regarding novel architectures, we believe the answer is "yes" -- our method does not make assumptions about model internals, so it remains applicable regardless of architecture.
>
> Regarding tasks with no prior data, IRT-based methods require some initial evaluation data. For a new task, this means benchmarking several models first to estimate item parameters.
>
> We agree that these are important considerations in practice, and we will make sure to mention them in the final version of the paper.

---

### Decision · Program_Chairs · 2025-07-08

**Decision:**

Accept

**Comment:**

- **Summary**: This paper proposes Fluid Benchmarking that dynamically selects evaluation items based on Item Response Theory (IRT) to address current challenges in LLM Benchmarking: efficiency, validity, variance, and saturation. This paper empirically demonstrated by showing consistent improvements over static and IRT-based baselines.
- **Review summary**: All reviewers leaned toward acceptance, praising the critical problem and the empirical results.
- **Strengths**:
(1) this paper addresses an important benchmarking issues;
(2) the idea is intuitive and grounded in theoretical foundation;
(3) it is empirically demonstrated;
(4) the paper is well-written.
- **Weakness**:
(1) the validity for models with heterogeneous capabilities -- additional analysis and discussion is needed;
(2) lack of full-benchmark evaluation;
(3) lack of discussion on generalization beyond train models and no prior data settings.
- **Meta-review decision**:
I recommend acceptance, as the paper deals a critical problem with proposing empirical solid methods and the reviewers’ concerns are addressable.